# On Controllable Sparse Alternatives to Softmax

**Anirban Laha**[1][†][*]    **Saneem A. Chemmengath**[1][*]    **Priyanka Agrawal**[1]    **Mitesh M. Khapra**[2]
**Karthik Sankaranarayanan**[1]        **Harish G. Ramaswamy**[2]
[1] IBM Research    [2] Robert Bosch Center for DS and AI, and Dept of CSE, IIT Madras

## Abstract

Converting an n-dimensional vector to a probability distribution over n objects is a commonly used component in many machine learning tasks like multiclass classification, multilabel classification, attention mechanisms etc. For this, several probability mapping functions have been proposed and employed in literature such as softmax, sum-normalization, spherical softmax, and sparsemax, but there is very little understanding in terms how they relate with each other. Further, none of the above formulations offer an explicit control over the degree of sparsity. To address this, we develop a unified framework that encompasses all these formulations as special cases. This framework ensures simple closed-form solutions and existence of sub-gradients suitable for learning via backpropagation. Within this framework, we propose two novel sparse formulations, *sparsegen-lin* and *sparsehourglass*, that seek to provide a control over the degree of desired sparsity. We further develop novel convex loss functions that help induce the behavior of aforementioned formulations in the multilabel classification setting, showing improved performance. We also demonstrate empirically that the proposed formulations, when used to compute attention weights, achieve better or comparable performance on standard seq2seq tasks like neural machine translation and abstractive summarization.

## 1   Introduction

Various widely used probability mapping functions such as sum-normalization, softmax, and spherical softmax enable mapping of vectors from the euclidean space to probability distributions. The need for such functions arises in multiple problem settings like multiclass classification [1, 2], reinforcement learning [3, 4] and more recently in attention mechanism [5, 6, 7, 8, 9] in deep neural networks, amongst others. Even though softmax is the most prevalent approach amongst them, it has a shortcoming in that its outputs are composed of only non-zeroes and is therefore ill-suited for producing sparse probability distributions as output. The need for sparsity is motivated by parsimonious representations [10] investigated in the context of variable or feature selection. Sparsity in the input space offers benefits of model interpretability as well as computational benefits whereas on the output side, it helps in filtering large output spaces, for example in large scale multilabel classification settings [11]. While there have been several such mapping functions proposed in literature such as softmax [4], spherical softmax [12, 13] and sparsemax [14, 15], very little is understood in terms of how they relate to each other and their theoretical underpinnings. Further, for sparse formulations, often there is a need to trade-off interpretability for accuracy, yet none of these formulations offer an explicit control over the desired degree of sparsity.

Motivated by these shortcomings, in this paper, we introduce a general formulation encompassing all such probability mapping functions which serves as a unifying framework to understand individual formulations such as hardmax, softmax, sum-normalization, spherical softmax and sparsemax as special cases, while at the same time helps in providing explicit control over degree of sparsity. With the

---

[*]Equal contribution by the first two authors. Corresponding authors: {anirlaha,saneem.cg}@in.ibm.com.
[†]This author was also briefly associated with IIT Madras during the course of this work.

aim of controlling sparsity, we propose two new formulations: **sparsegen-lin** and **sparsehourglass**. Our framework also ensures simple closed-form solutions and existence of sub-gradients similar to softmax. This enables them to be employed as activation functions in neural networks which require gradients for backpropagation and are suitable for tasks that require sparse attention mechanism [14]. We also show that the **sparsehourglass** formulation can extend from translation invariance to scale invariance with an explicit control, thus helping to achieve an adaptive trade-off between these invariance properties as may be required in a problem domain.

We further propose new convex loss functions which can help induce the behaviour of the above proposed formulations in a multilabel classification setting. These loss functions are derived from a violation of constraints required to be satisfied by the corresponding mapping functions. This way of defining losses leads to an alternative loss definition for even the sparsemax function [14]. Through experiments we are able to achieve improved results in terms of sparsity and prediction accuracies for multilabel classification.

Lastly, the existence of sub-gradients for our proposed formulations enable us to employ them to compute attention weights [5, 7] in natural language generation tasks. The explicit controls provided by sparsegen-lin and sparsehourglass help to achieve higher interpretability while providing better or comparable accuracy scores. A recent work [16] had also proposed a framework for attention; however, they had not explored the effect of explicit sparsity controls. To summarize, our contributions are the following:

- A general framework of formulations producing probability distributions with connections to hardmax, softmax, sparsemax, spherical softmax and sum-normalization (*Sec.*3).
- New formulations like **sparsegen-lin** and **sparsehourglass** as special cases of the general framework which enable explicit control over the desired degree of sparsity (*Sec.*3.2,3.5).
- A formulation **sparsehourglass** which enables us to adaptively trade-off between the translation and scale invariance properties through explicit control (*Sec.*3.5).
- Convex multilabel loss functions correponding to all the above formulations proposed by us. This enable us to achieve improvements in the multilabel classification problem (*Sec.*4).
- Experiments for sparse attention on natural language generation tasks showing comparable or better accuracy scores while achieving higher interpretability (*Sec.*5).

## 2   Preliminaries and Problem Setup

**Notations:** For $K \in \mathbb{Z}_+$, we denote $[K] := \{1, \ldots, K\}$. Let $\boldsymbol{z} \in \mathbb{R}^K$ be a real vector denoted as $\boldsymbol{z} = \{z_1, \ldots, z_K\}$. $\mathbf{1}$ and $\mathbf{0}$ denote vector of ones and zeros resp. Let $\Delta^{K-1} := \{\boldsymbol{p} \in \mathbb{R}^K \mid \mathbf{1}^T \boldsymbol{p} = 1, \boldsymbol{p} \geq \mathbf{0}\}$ be the $(K-1)$-dimensional simplex and $\boldsymbol{p} \in \Delta^{K-1}$ be denoted as $\boldsymbol{p} = \{p_1, \ldots, p_K\}$. We use $[t]_+ := \max\{0, t\}$. Let $A(\boldsymbol{z}) := \{k \in [K] \mid z_k = \max_j z_j\}$ be the set of maximal elements of $z$.

**Definition**: A *probability mapping function* is a map $\rho : \mathbb{R}^K \to \Delta^{K-1}$ which transforms a score vector $\boldsymbol{z}$ to a categorical distribution (denoted as $\rho(\boldsymbol{z}) = \{\rho_1(\boldsymbol{z}), \ldots, \rho_K(\boldsymbol{z})\}$). The support of $\rho(\boldsymbol{z})$ is $S(\boldsymbol{z}) := \{j \in [K] \mid \rho_j(\boldsymbol{z}) > 0\}$. Such mapping functions can be used as activation function for machine learning models. Some known probability mapping functions are listed below:

- **Softmax** function is defined as: $\rho_i(\boldsymbol{z}) = \frac{\exp(z_i)}{\sum_{j \in [K]} \exp(z_j)}, \ \forall i \in [K]$. Softmax is easy to evaluate and differentiate and its logarithm is the negative log-likelihood loss [14].
- **Spherical softmax** - Another function which is simple to compute and derivative-friendly: $\rho_i(\boldsymbol{z}) = \frac{z_i^2}{\sum_{j \in [K]} z_j^2}, \ \forall i \in [K]$. Spherical softmax is not defined for $\sum_{j \in [K]} z_j^2 = 0$.
- **Sum-normalization** : $\rho_i(\boldsymbol{z}) = \frac{z_i}{\sum_{j \in [K]} z_j}, \ \forall i \in [K]$. It is not used in practice much as the mapping is not defined if $z_i < 0$ for any $i \in [K]$ and for $\sum_{j \in [K]} z_j = 0$.

The above mapping functions are limited to producing distributions with full support. Consider there is a single value of $z_i$ significantly higher than the rest, its desired probability should be exactly 1, while the rest should be grounded to zero (*hardmax* mapping). Unfortunately, that does not happen unless the rest of the values tend to $-\infty$ (in case of softmax) or are equal to 0 (in case of spherical softmax and sum-normalization).

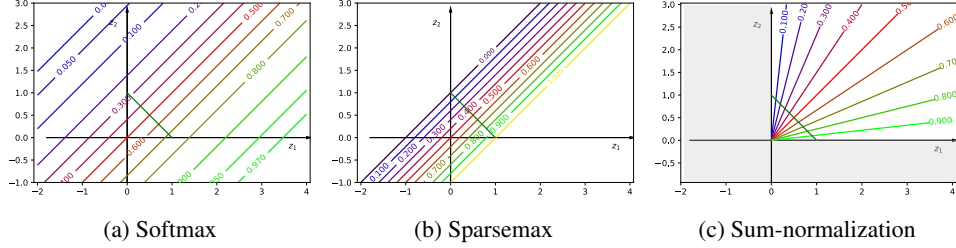

|  (a) Softmax | (b) Sparsemax | (c) Sum-normalization |

Figure 1: **Visualization of probability mapping functions** in two-dimension. The contour plots show values of $\rho_1(z)$. The green line segment connecting (1,0) and (0,1) is the 1-dimensional probability simplex. Each contour (here it is line) contains points in $\mathbb{R}^2$ plane which have the same $\rho_1(z)$, the exact value marked on the contour line.

- **Sparsemax** recently introduced by [14] circumvents this issue by projecting the score vector $z$ onto a simplex [15]: $\rho(z) = \operatorname{argmin}_{p \in \Delta^{K-1}} \|p - z\|_2^2$. This offers an intermediate solution between softmax (no zeroes) and hardmax (zeroes except for the highest value).

The contour plots for softmax, sparsemax and sum-normalization in two-dimensions ($z \in \mathbb{R}^2$) are shown in *Fig.*1. The contours of sparsemax are concentrated over a narrow region, while the remaining region corresponds to sparse solutions. For softmax, the contour plots are spread over the whole real plane, confirming the absence of sparse solutions. Sum-normalization is not defined outside the first quadrant, and yet, the contours cover the whole quadrant, denying sparse solutions.

## 3   Sparsegen Activation Framework

**Definition**: We propose a generic probability mapping function inspired from the sparsemax formulation (in *Sec.*2) which we call **sparsegen**:

$$\rho(z) = \operatorname{sparsegen}(z; g, \lambda) = \operatorname*{argmin}_{p \in \Delta^{K-1}} \|p - g(z)\|_2^2 - \lambda\|p\|_2^2 \tag{1}$$

where $g : \mathbb{R}^K \to \mathbb{R}^K$ is a component-wise *transformation function* applied on $z$. Here $g_i(z)$ denotes the $i$-th component of $g(z)$. The coefficient $\lambda < 1$ controls the regularization strength. For $\lambda > 0$, the second term becomes *negative L-2 norm* of $p$. In addition to minimizing the error on projection of $g(z)$, *Eq.*1 tries to maximize the norm, which encourages larger probability values for some indices, hence moving the rest to zero. The above formulation has a closed-form solution (see App. A.1 for solution details), which can be computed in $O(K)$ time using the modified randomized median finding algorithm as followed in [15] while solving the *projection onto simplex* problem.

The choices of both $\lambda$ and $g$ can help control the cardinality of the support set $S(z)$, thus influencing the sparsity of $\rho(z)$. $\lambda$ can help produce distributions with support ranging from full (*uniform distribution* when $\lambda \to 1^-$) to minimum (*hardmax* when $\lambda \to -\infty$). Let $S(z, \lambda_1)$ denote the support of sparsegen for a particular coefficient $\lambda_1$. It is easy to show: if $|S(z, \lambda_1)| > |A(z)|$, then there exists $\lambda_x > \lambda_1$ for an $x < |S(z, \lambda_1)|$ such that $|S(z, \lambda_x)| = x$. In other words, if a sparser solution exists, it can be obtained by changing $\lambda$. The following result has an alternate interpretation for $\lambda$:

**Result**: *The **sparsegen** formulation (Eq.1) is equivalent to the following, when $\gamma = \frac{1}{1-\lambda}$ (where $\gamma > 0$):* $\rho(z) = \operatorname{argmin}_{p \in \Delta^{K-1}} \|p - \gamma g(z)\|_2^2$.

The above result says that scaling $g(z)$ by $\gamma = \frac{1}{1-\lambda}$ is equivalent to applying the negative $L$-2 norm with $\lambda$ coefficient when considering projection of $g(z)$ onto the simplex. Thus, we can write:

$$\operatorname{sparsegen}(z; g, \lambda) = \operatorname{sparsemax}\left(\frac{g(z)}{1-\lambda}\right). \tag{2}$$

This equivalence helps us borrow results from sparsemax to establish various properties for sparsegen.

**Jacobian of sparsegen**: To train a model with sparsegen as an activation function, it is essential to compute its *Jacobian* matrix denoted by $J_\rho(z) := [\partial\rho_i(z)/\partial z_j]_{i,j}$ for using gradient-based

optimization techniques. We use *Eq.*2 and results from [14](*Sec.*2.5) to derive the Jacobian for sparsegen by applying chain rule of derivatives:

$$J_{\text{sparsegen}}(\boldsymbol{z}) = J_{\text{sparsemax}}\left(\frac{g(\boldsymbol{z})}{1-\lambda}\right) \times \frac{J_g(\boldsymbol{z})}{1-\lambda} \tag{3}$$

where $J_g(\boldsymbol{z})$ is Jacobian of $g(\boldsymbol{z})$ and $J_{\text{sparsemax}}(\boldsymbol{z}) = \left[\text{Diag}(\boldsymbol{s}) - \frac{\boldsymbol{s}\boldsymbol{s}^T}{|S(\boldsymbol{z})|}\right]$. Here $\boldsymbol{s}$ is an indicator vector whose $i^{\text{th}}$ entry is 1 if $i \in S(\boldsymbol{z})$. $\text{Diag}(\boldsymbol{s})$ is a matrix created using $\boldsymbol{s}$ as its diagonal entries.

## 3.1 Special cases of Sparsegen: sparsemax, softmax and spherical softmax

Apart from $\lambda$, one can control the sparsity of sparsegen through $g(\boldsymbol{z})$ as well. Moreover, certain choices of $\lambda$ and $g(\boldsymbol{z})$ help us establish connections with existing activation functions (see *Sec.*2). The following cases illustrate these connections (more details in *App.*A.2):

**Example 1**: $g(\boldsymbol{z}) = \exp(\boldsymbol{z})$ (**sparsegen-exp**): $\exp(\boldsymbol{z})$ denotes element-wise exponentiation of $\boldsymbol{z}$, that is $g_i(\boldsymbol{z}) = \exp(z_i)$. Sparsegen-exp reduces to softmax when $\lambda = 1 - \sum_{j\in[K]} e^{z_j}$, as it results in $S(\boldsymbol{z}) = [K]$ as per *Eq.*14 in *App.*A.2.

**Example 2**: $g(\boldsymbol{z}) = \boldsymbol{z}^2$ (**sparsegen-sq**): $\boldsymbol{z}^2$ denotes element-wise square of $\boldsymbol{z}$. As observed for sparsegen-exp, when $\lambda = 1 - \sum_{j\in[K]} z_j^2$, sparsegen-sq reduces to spherical softmax.

**Example 3** : $g(\boldsymbol{z}) = \boldsymbol{z}, \lambda = 0$: This case is equivalent to the projection onto the simplex objective adopted by sparsemax. Setting $\lambda \neq 0$ leads the regularized extension of sparsemax as seen next.

## 3.2 Sparsegen-lin: Extension of sparsemax

The negative $L$-2 norm regularizer in *Eq.*4 helps to control the width of the non-sparse region (see *Fig.*2 for the region plot in two-dimensions). In the extreme case of $\lambda \to 1^-$, the whole real plane maps to sparse region whereas for $\lambda \to -\infty$, the whole real plane renders non-sparse solutions.

$$\rho(\boldsymbol{z}) = \text{sparsegen-lin}(\boldsymbol{z}) = \underset{\boldsymbol{p}\in\Delta^{K-1}}{\text{argmin}} \ \|\boldsymbol{p}-\boldsymbol{z}\|_2^2 - \lambda\|\boldsymbol{p}\|_2^2 \tag{4}$$

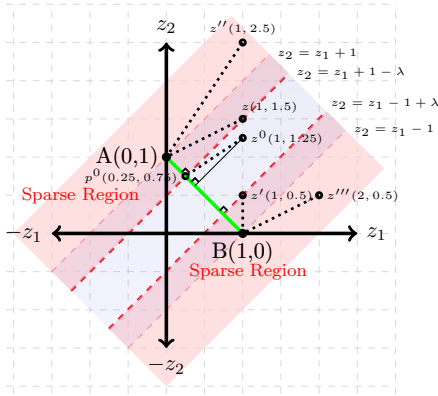

Figure 2: **Sparsegen-lin**: Region plot for $\boldsymbol{z} = \{z_1, z_2\} \in \mathbb{R}^2$ when $\lambda = 0.5$. $\rho(\boldsymbol{z})$ is sparse in the red region, whereas non-sparse in the blue region. The dashed red lines depict the boundaries between sparse and non-sparse regions. For $\lambda = 0.5$, points like ($\boldsymbol{z}$ or $\boldsymbol{z}'$) are mapped onto the sparse points **A** or **B**. Whereas for sparsemax ($\lambda = 0$), they fall in the blue region (the boundaries of sparsemax are shown by lighter dashed red lines passing through **A** and **B**). The point $\boldsymbol{z}^0$ lies in the blue region, producing non-sparse solution. Interestingly more points like $\boldsymbol{z}''$ and $\boldsymbol{z}'''$, which currently lie in the red region can fall in the blue region for some $\lambda < 0$. For $\lambda > 0.5$, the blue region becomes smaller, as mores points map to sparse solutions.

## 3.3 Desirable properties for probability mapping functions

Let us enumerate below some properties a probability mapping function $\rho$ should possess:

1. *Monotonicity*: If $z_i \geq z_j$, then $\rho_i(\boldsymbol{z}) \geq \rho_j(\boldsymbol{z})$. This does not always hold true for sum-normalization and spherical softmax when one or both of $z_i, z_j$ is less than zero. For sparsegen, both $g_i(\boldsymbol{z})$ and $g_i(\boldsymbol{z})/(1-\lambda)$ should be monotonic increasing, which implies $\lambda$ needs to be less than 1.

2. *Full domain*: The domain of $\rho$ should include negatives as well as positives, i.e. $\text{Dom}(\rho) \in \mathbb{R}^K$. Sum-normalization does not satisfy this as it is not defined if some dimensions of $\boldsymbol{z}$ are negative.

3. *Existence of Jacobian*: This enables usage in any training algorithm where gradient-based optimization is used. For sparsegen, the Jacobian of $g(\boldsymbol{z})$ should be easily computable (*Eq.*3).

4. *Lipschitz continuity*: The derivative of the function should be upper bounded. This is important for the stability of optimization technique used in training. Softmax and sparsemax are 1-Lipschitz whereas spherical softmax and sum-normalization are not Lipschitz continuous. *Eq.*3 shows the Lipschitz constant for sparsegen is upper bounded by $1/(1-\lambda)$ times the Lipschitz constant for $g(\boldsymbol{z})$.

5. *Translation invariance*: Adding a constant $c$ to every element in $\boldsymbol{z}$ should not change the output distribution : $\rho(\boldsymbol{z} + c\mathbf{1}) = \rho(\boldsymbol{z})$. Sparsemax and softmax are translation invariant whereas sum-normalization and spherical softmax are not. Sparsegen is translation invariant iff for all $c \in \mathbb{R}$ there exist a $\tilde{c} \in \mathbb{R}$ such that $g(\boldsymbol{z} + c\mathbf{1}) = g(\boldsymbol{z}) + \tilde{c}\mathbf{1}$. This follows from *Eq.*2.

6. *Scale invariance*: Multiplying every element in $\boldsymbol{z}$ by a constant $c$ should not change the output distribution : $\rho(c\boldsymbol{z}) = \rho(\boldsymbol{z})$. Sum-normalization and spherical softmax satisfy this property whereas sparsemax and softmax are not scale invariant. Sparsegen is scale invariant iff for all $c \in \mathbb{R}$ there exist a $\hat{c} \in \mathbb{R}$ such that $g(c\boldsymbol{z}) = g(\boldsymbol{z}) + \hat{c}\mathbf{1}$. This also follows from *Eq.*2.

7. *Permutation invariance*: If there is a permutation matrix $\boldsymbol{P}$, then $\rho(\boldsymbol{P}\boldsymbol{z}) = \boldsymbol{P}\rho(\boldsymbol{z})$. For sparsegen, the precondition is that $g(\boldsymbol{z})$ should be a permutation invariant function.

8. *Idempotence*: $\rho(\boldsymbol{z}) = \boldsymbol{z}, \forall \boldsymbol{z} \in \Delta^{K-1}$. This is true for sparsemax and sum-normalization. For sparsegen, it is true if and only if $g(\boldsymbol{z}) = \boldsymbol{z}, \forall \boldsymbol{z} \in \Delta^{K-1}$ and $\lambda = 0$.

In the next section, we discuss in detail about the scale invariance and translation invariance properties and propose a new formulation achieving a trade-off between these properties.

## 3.4   Trading off Translation and Scale Invariances

As mentioned in *Sec.*1, scale invariance is a desirable property to have for probability mapping functions. Consider applying sparsemax on two vectors $\boldsymbol{z} = \{0, 1\}$, $\bar{\boldsymbol{z}} = \{100, 101\} \in \mathbb{R}^2$. Both would result in $\{0, 1\}$ as the output. However, ideally $\bar{\boldsymbol{z}}$ should have mapped to a distribution nearer to $\{0.5, 0.5\}$ instead. Scale invariant functions will not have such a problem. Among the existing functions, only sum-normalization and spherical softmax satisfy scale invariance. While sum-normalization is only defined for positive values of $\boldsymbol{z}$, spherical softmax is not monotonic or Lipschitz continuous. In addition, both of these methods are also not defined for $\boldsymbol{z} = \mathbf{0}$, thus making them unusable for practical purposes. It can be shown that any probability mapping function with the scale invariance property will not be Lipschitz continuous and will be undefined for $\boldsymbol{z} = \mathbf{0}$.

A recent work[13] had pointed out the lack of clarity over whether scale invariance is more desired than the translation invariance property of softmax and sparsemax. We take this into account to achieve trade-off between the two invariances. In the usual scale invariance property, scaling vector $\boldsymbol{z}$ essentially results in another vector along the line connecting $\boldsymbol{z}$ and the origin. That resultant vector also has the same output probability distribution as the original vector (See *Sec.* 3.3). We propose to scale the vector $\boldsymbol{z}$ along the line connecting it with a point (we call it *anchor point* henceforth) other than the origin, yet achieving the same output. Interestingly, the choice of this anchor point can act as a control to help achieve a trade-off between scale invariance and translation invariance.

Let a vector $\boldsymbol{z}$ be projected onto the simplex along the line connecting it with an *anchor point* $\boldsymbol{q} = (-q, \ldots, -q) \in \mathbb{R}^K$, for $q > 0$ (See *Fig.*3a for $K = 2$). We choose $g(\boldsymbol{z})$ as the point where this line intersects with the affine hyperplane $\mathbf{1}^T \hat{\boldsymbol{z}} = 1$ containing the probability simplex. Thus, $g(\boldsymbol{z})$ is set equal to $\alpha \boldsymbol{z} + (1 - \alpha)\boldsymbol{q}$, where $\alpha = \frac{1 + Kq}{\sum_i z_i + Kq}$ (we denote it as $\alpha(\boldsymbol{z})$ as $\alpha$ is a function of $\boldsymbol{z}$). From the translation invariance property of sparsemax, the resultant mapping function can be shown equivalent to considering $g(\boldsymbol{z}) = \alpha(\boldsymbol{z})\boldsymbol{z}$ in *Eq.*1. We refer to this variant of sparsegen assuming $g(\boldsymbol{z}) = \alpha(\boldsymbol{z})\boldsymbol{z}$ and $\lambda = 0$ as *sparsecone*.

Interestingly, when the parameter $q = 0$, sparsecone reduces to sum-normalization (scale invariant) and when $q \to \infty$, it is equivalent to sparsemax (translation invariant). Thus the parameter $q$ acts as a control taking sparsecone from scale invariance to translation invariance. At intermediate values (that is, for $0 < q < \infty$), sparsecone is *approximate scale invariant* with respect to the anchor point $\boldsymbol{q}$. However, it is undefined for $\boldsymbol{z}$ where $\sum_i z_i < -Kq$ (beyond the black dashed line shown in *Fig.*3a). In this case the denominator term of $\alpha(\boldsymbol{z})$ (that is, $\sum_i z_i + Kq$) becomes negative destroying the monotonicity of $\alpha(\boldsymbol{z})\boldsymbol{z}$. Also note that sparsecone is not Lipschitz continuous.

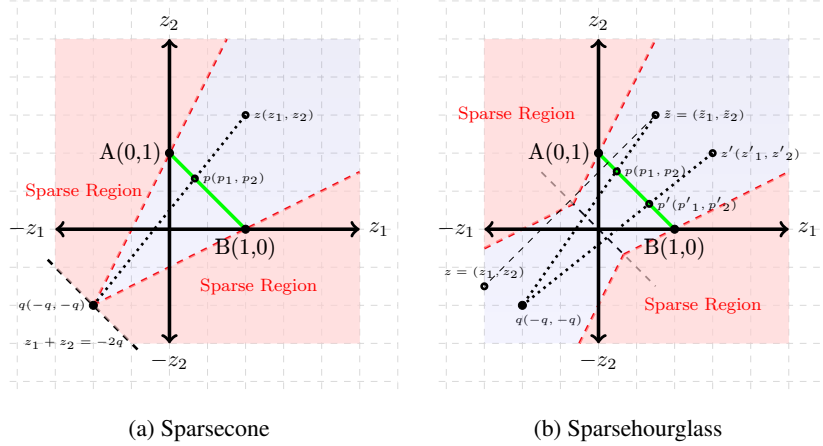

Figure 3: (a) **Sparsecone:** The vector $z$ maps to a point $p$ on the simplex along the line connecting to the point $q = (-q, -q)$. Here we consider $q = 1$. The red region corresponds to sparse region whereas blue covers the non-sparse region. (b) **Sparsehourglass:** For the vector $z'$ in the positive half-space, the mapping to the solution $p'$ can be obtained similarly as sparsecone. For the vector $z$ in the negative half-space, a mirror point $\tilde{z}$ needs to be found, which leads to the solution $p$.

### 3.5 Proposed Solution: Sparsehourglass

To alleviate the issue of monotonicity when $\sum_i z_i < -Kq$, we choose to restrict applying sparsecone only for the positive half-space $\mathbb{H}^K_+ := \{z \in \mathbb{R}^K \mid \sum_i z_i \geq 0\}$. For the remaining negative half-space $\mathbb{H}^K_- := \{z \in \mathbb{R}^K \mid \sum_i z_i < 0\}$, we define a **mirror point function** to transform to a point in $\mathbb{H}^K_+$, on which sparsecone can be applied. Thus the solution for a point in the negative half-space is given by the solution of its corresponding mirror point in the positive half-space. This mirror point function has some necessary properties (see App. A.4 for details), which can be satisfied by defining $m$: $m_i(z) = z_i - \frac{2 \sum_j z_j}{K}, \forall i \in [K]$. Interestingly, this can alternatively be achieved by choosing $g(z) = \hat{\alpha}(z)z$, where $\hat{\alpha}$ is a slight modification of $\alpha$ given by $\hat{\alpha}(z) = \frac{1+Kq}{|\sum_i z_i|+Kq}$. This leads to the definition of a new probability mapping function (which we call *sparsehourglass*):

$$\rho(z) = \text{sparsehourglass}(z) = \underset{p \in \Delta^{K-1}}{\text{argmin}} \left\| p - \frac{1+Kq}{|\sum_{i \in [K]} z_i| + Kq} z \right\|_2^2 \tag{5}$$

Like sparsecone, sparsehourglass also reduces to sparsemax when $q \to \infty$. Similarly, $q = 0$ for sparsehourglass leads to a corrected version of sum-normalization (we call it *sum normalization++* ), which works for the negative domain as well unlike the original version defined in *Sec.*2. Another advantage of sparsehourglass is that it is Lipschitz continuous with Lipschitz constant equal to $(1 + \frac{1}{Kq})$ (proof details in *App.*A.5). *Table.*1 summarizes all the formulations seen in this paper and compares them against the various important properties mentioned in *Sec.*3.3. Note that **sparsehourglass is the only probability mapping function which satisfies all the properties**. Even though it does not satisfy both scale invariance and translation invariance simultaneously, it is possible to achieve these separately through different values of $q$ parameter, which can be decided independent of $z$.

## 4 Sparsity Inducing Loss Functions for Multilabel Classification

An important usage of such sparse probability mapping functions is in the output mapping of multilabel classification models. Typical multilabel problems have hundreds of possible labels or tags, but any single instance has only a few tags [17]. Thus, a function which takes in a vector in $\mathbb{R}^K$ and outputs a sparse version of the vector is of great value.

Given training instances $(x_i, y_i) \in \mathcal{X} \times \{0, 1\}^K$, we need to find a model function $f : \mathcal{X} \to \mathbb{R}^K$ that produces score vector over the label space, which on application of $\rho : \mathbb{R}^K \to \Delta^{K-1}$ (the sparse probability mapping function in question) leads to correct prediction of label vector $y_i$. Define

Table 1: Summary of the properties satisfied by probability mapping functions. Here ✔ denotes 'satisfied in general', ✗ signifies 'not satisfied' and √ says 'satisfied for some constant parameter independent of $z$'. Note that PERMUTATION INV and existence of JACOBIAN are satisfied by all.

| FUNCTION | IDEMPOTENCE | MONOTONIC | TRANSLATION INV | SCALE INV | FULL DOMAIN | LIPSCHITZ |
|---|---|---|---|---|---|---|
| SUM NORMALIZATION | ✔ | ✗ | ✗ | ✔ | ✗ | $\infty$ |
| SPHERICAL SOFTMAX | ✗ | ✗ | ✗ | ✔ | ✗ | $\infty$ |
| SOFTMAX | ✗ | ✔ | ✔ | ✗ | ✔ | 1 |
| SPARSEMAX | ✔ | ✔ | ✔ | ✗ | ✔ | 1 |
| SPARSEGEN-LIN | √ | ✔ | ✔ | ✗ | ✔ | $1/(1-\lambda)$ |
| SPARSEGEN-EXP | ✗ | ✔ | ✗ | ✗ | ✔ | $\infty$ |
| SPARSEGEN-SQ | ✗ | ✗ | ✗ | ✗ | ✔ | $\infty$ |
| SPARSECONE | ✔ | ✗ | √ | √ | ✗ | $\infty$ |
| **SPARSEHOURGLASS** | ✔ | ✔ | √ | √ | ✔ | $(1 + 1/Kq)$ |
| SUM NORMALIZATION++ | ✔ | ✔ | ✗ | ✔ | ✔ | $\infty$ |

$\boldsymbol{\eta}_i := \boldsymbol{y}_i / \|\boldsymbol{y}_i\|_1$, which is a probability distribution over the labels. Considering a loss function $\mathscr{L} : \Delta^{K-1} \times \Delta^{K-1} \to [0, \infty)$ and representing $\boldsymbol{z}_i := f(\boldsymbol{x}_i)$, a natural way for training using $\rho$ is to find a function $f : \mathcal{X} \to \mathbb{R}^K$ that minimises the error $R(f)$ below over a hypothesis class $\mathcal{F}$:

$$R(f) = \sum_{i=1}^{M} \mathscr{L}(\rho(\boldsymbol{z}_i), \boldsymbol{\eta}_i) \tag{6}$$

In the prediction phase, for a test instance $\boldsymbol{x}$, one can simply predict the non-zero elements in the vector $\rho(f^*(\boldsymbol{x}))$ where $f^*$ is the minimizer of the above training objective $R(f)$.

For all cases where $\rho$ produces sparse probability distributions, one can show that the training objective $R$ above is highly non-convex in $f$, even for the case of a linear hypothesis class $\mathcal{F}$. However, if we remove the strict requirement of the training objective depending on $\rho(\boldsymbol{z})$ (as in *Eq.*6), and use a loss function which can work with $\boldsymbol{z}$ directly, a convex objective is possible. We, thus, design a loss function $\mathcal{L} : \mathbb{R}^K \times \Delta^{K-1} \to [0, \infty)$ such that $\mathcal{L}(\boldsymbol{z}, \boldsymbol{\eta}) = 0$ only if $\rho(\boldsymbol{z}) = \boldsymbol{\eta}$. To derive such a loss function, we proceed by enumerating a list of constraints which will be satisfied by the zero-loss region in the $K$-dimensional space of the vector $\boldsymbol{z}$. For sparsehourglass, the closed-form solution is given by $\rho_i(\boldsymbol{z}) = [\hat{\alpha}(\boldsymbol{z})z_i - \tau(\boldsymbol{z})]_+$ (see *App.*A.1). This enables us to list down the following constraints for zero loss: (1) $\hat{\alpha}(\boldsymbol{z})(z_i - z_j) = 0$, $\forall i, j \, | \eta_i = \eta_j \neq 0$, and (2) $\hat{\alpha}(\boldsymbol{z})(z_i - z_j) \geq \eta_i$, $\forall i, j \, | \eta_i \neq 0, \eta_j = 0$. The value of the loss when any such constraints is violated is simply determined by piece-wise linear functions, which lead to the following loss function for sparsehourglass:

$$\mathcal{L}_{\text{sparsehg,hinge}}(\boldsymbol{z}, \boldsymbol{\eta}) = \sum_{\substack{i,j \\ \eta_i \neq 0, \eta_j \neq 0}} |z_i - z_j| + \sum_{\substack{i,j \\ \eta_i \neq 0, \eta_j = 0}} \max\left\{ \frac{\eta_i}{\hat{\alpha}(\boldsymbol{z})} - (z_i - z_j), 0 \right\}. \tag{7}$$

It can be easily proved that the above loss function is convex in $\boldsymbol{z}$ using the properties that both sum of convex functions and maximum of convex functions result in convex functions. The above strategy can also be applied to derive a multilabel loss function for sparsegen-lin:

$$\mathcal{L}_{\text{sparsegen-lin,hinge}}(\boldsymbol{z}, \boldsymbol{\eta}) = \frac{1}{1-\lambda} \sum_{\substack{i,j \\ \eta_i \neq 0, \eta_j \neq 0}} |z_i - z_j| + \sum_{\substack{i,j \\ \eta_i \neq 0, \eta_j = 0}} \max\left\{ \eta_i - \frac{z_i - z_j}{1-\lambda}, 0 \right\}. \tag{8}$$

The above loss function for sparsegen-lin can be used to derive a multilabel loss for sparsemax by setting $\lambda = 0$ (which we use in our experiments for "sparsemax+hinge") . The piecewise-linear losses proposed in this section based on violation of constraints are similar to the well-known hinge loss, whereas the sparsemax loss proposed by [14] (which we use in our experiments for "sparsemax+huber") has connections with Huber loss. We have shown through our experiments in next section, that hinge loss variants for multilabel classification work better than Huber loss variants.

## 5 Experiments and Results

Here we present two sets of evaluations for the proposed probability mapping functions and loss functions. First, we apply them on the multilabel classification task studying the effect of varying label density in synthetic dataset, followed by evaluation on real multilabel datasets. Next, we report results of sparse attention on NLP tasks of machine translation and abstractive summarization.

## 5.1 Multilabel Classification

We compare the proposed activations and loss functions for multilabel classification with both synthetic and real datasets. We use a linear prediction model followed by a loss function during training. During test time, the corresponding activation is directly applied to the output of the linear model. We consider the following activation-loss pairs: (1) **softmax+log**: KL-divergence loss applied on top of softmax outputs, (2) **sparsemax+huber**: multilabel classification method from [14], (3) **sparsemax+hinge**: hinge loss as in *Eq.*8 with $\lambda = 0$ is used during training compared to Huber loss in (2), and (4) **sparsehg+hinge**: for sparsehourglass (in short sparsehg), loss in *Eq.*7 is used during training. Please note as we have a convex system of equations due to an underlying linear prediction model, applying *Eq.*8 in training and applying sparsegen-lin activation during test time produces the same result as sparsemax+hinge. For softmax+log, we used a threshold $p_0$, above which a label is predicted to be "on". For others, a label is predicted "on" if its predicted probability is non-zero. We tune hyperparams $q$ for sparsehg+hinge and $p_0$ for softmax+log using validation set.

### 5.1.1 Synthetic dataset with varied label density

We use scikit-learn for generating synthetic datasets (details in *App.*A.6). We conducted experiments in **three settings**: (1) varying mean number of labels per instance, (2) varying range of number of labels and, (3) varying document length. In the first setting, we study the ability to model varying label sparsity. We draw number of labels $N$ uniformly at random from set $\{\mu - 1, \mu, \mu + 1\}$ where $\mu \in \{2 \ldots 9\}$ is mean number of labels. For the second setting we study how these models perform when label density varies across instances. We draw $N$ uniformly at random from set $\{5 - r, \ldots, 5 + r\}$. Parameter $r$ controls variation of label density among instances. In the third setting we experiment with different document lengths, we draw $N$ from Poisson with mean 5 and vary document length $L$ from 200 to 2000. In first two settings document length was fixed to 2000.

We report F-score[2] and Jensen-Shannon divergence (JSD) on test set in our results.

*Fig.*5 shows F-score on test sets in the three experimental settings. We can observe that sparsemax+hinge and sparsehg+hinge consistently perform better than sparsemax+huber in all three cases, especially the label distributions are sparser. Note that sparsehg+hinge performs better than sparsemax+hinge in most cases. From empirical comparison between sparsemax+hinge and sparsemax+huber, we can conclude that the proposed hinge loss variants are better in producing sparser and and more accurate predictions. This observation is also supported in our analysis of sparsity in outputs (see *Fig.*4 - lower the curve the sparser it is - this is analysis is done corresponding to the setting in *Fig.*5a), where we find that hinge

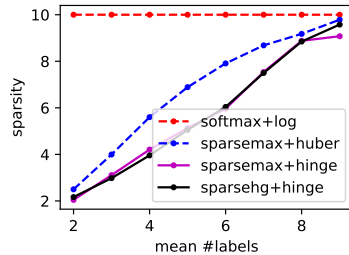

Figure 4: Sparsity comparison

loss variants encourage more sparsity. We also find the hinge loss variants are doing better than softmax+log in terms of the JSD metric (details in *App.*A.7.1).

### 5.1.2 Real Multilabel datasets

We further experiment with three real datasets[3] for multilabel classification: Birds, Scene and Emotions. The experimental setup and baselines are same as that for synthetic dataset described in *Sec.*5.1.1. For each of the datasets, we consider only those examples with atleast one label. Results are shown in Table 3 in *App.*A.7.2. All methods give comparable results on these benchmark datasets.

## 5.2 Sparse Attention for Natural Language Generation

Here we demonstrate the effectiveness of our formulations experimentally on two natural language generation tasks: neural machine translation and abstractive sentence summarization. The purpose of these experiments are two fold: firstly, effectiveness of our proposed formulations *sparsegen-lin* and *sparsehourglass* in attention framework on these tasks, and secondly, control over sparsity leads to enhanced interpretability. We borrow the encoder-decoder architecture with attention (see *Fig.*7 in *App.*A.8). We replace the softmax function in attention by our proposed functions as well as

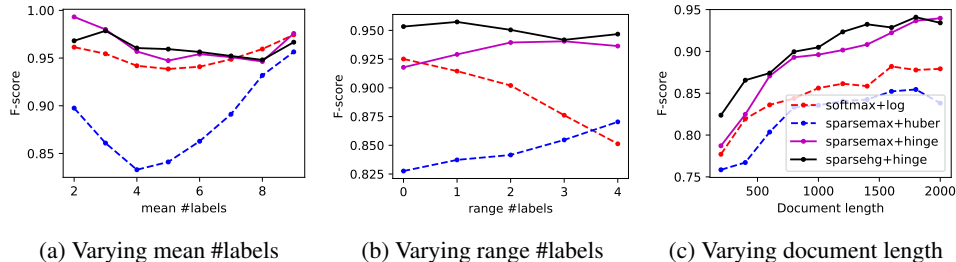

| (a) Varying mean #labels | (b) Varying range #labels | (c) Varying document length |

Figure 5: F-score on multilabel classification synthetic dataset.

Table 2: Sparse Attention Results. Here R-1, R-2 and R-L denote the ROUGE scores.

| | TRANSLATION | | SUMMARIZATION | | | | | | | | |
|---|---|---|---|---|---|---|---|---|---|---|---|
| Attention | FR-EN | EN-FR | Gigaword | | | DUC 2003 | | | DUC 2004 | | |
| | BLEU | BLEU | R-1 | R-2 | R-L | R-1 | R-2 | R-L | R-1 | R-2 | R-L |
| softmax | 36.38 | 36.00 | 34.80 | 16.64 | 32.15 | 27.95 | **9.22** | 24.54 | 30.68 | 12.24 | 28.12 |
| softmax (with temp.) | 36.63 | **36.08** | 35.00 | 17.15 | 32.57 | 27.78 | 8.91 | 24.53 | 31.64 | **12.89** | 28.51 |
| sparsemax | 36.73 | 35.78 | 34.89 | 16.88 | 32.20 | 27.29 | 8.48 | 24.04 | 30.80 | 12.01 | 28.04 |
| sparsegen-lin | **37.27** | 35.78 | **35.90** | **17.57** | **33.37** | **28.13** | 9.00 | **24.89** | **31.85** | 12.28 | **29.13** |
| sparsehg | 36.63 | 35.69 | 35.14 | 16.91 | 32.66 | 27.39 | 9.11 | 24.53 | 30.64 | 12.05 | 28.18 |

sparsemax as baseline. In addition we use another baseline where we tune for the temperature in softmax function. More details are provided in *App.*A.8.

**Experimental Setup**: In our experiments we adopt the same experimental setup followed by [16] on top of the OpenNMT framework [18]. We varied only the control parameters required by our formulations. The models for the different control parameters were trained for 13 epochs and the epoch with the best validation accuracy is chosen as the best model for that setting. The best control parameter for a formulation is again selected based on validation accuracy. For all our formulations, we report the test scores corresponding to the best control parameter in Table 2.

**Neural Machine Translation**: We consider the FR-EN language pair from the NMT-Benchmark project and perform experiments both ways. We see (refer *Table.*2) that sparsegen-lin surpasses BLEU scores of softmax and sparsemax for FR-EN translation, whereas sparsehg formulations yield comparable performance. Quantitatively, these metrics show that adding explicit controls do not come at the cost of accuracy. In addition, it is encouraging to see (refer *Fig.*8 in *App.*A.8) that increasing $\lambda$ for sparsegen-lin leads to crisper and hence more interpretable attention heatmaps (the lesser number of activated columns per row the better it is). We have also analyzed the average sparsity of heatmaps over the whole test dataset and have indeed observed that larger $\lambda$ leads to sparser attention.

**Abstractive Summarization**: We next perform our experiments on abstractive summarization datasets like Gigaword, DUC2003 & DUC2004 and report ROUGE metrics. The results in *Table.*2 show that sparsegen-lin stands out in performance with other formulations closely following and comparable to softmax and sparsemax. It is also encouraging to see that all the models trained on Gigaword generalizes well on other datasets DUC2003 and DUC2004. Here again the $\lambda$ control leads to more interpretable attention heatmaps as shown in *Fig.*9 in *App.*A.8 and we have also observed the same with average sparsity of heatmaps over the test set.

## 6 Conclusions and Future Work

In this paper, we investigated a family of sparse probability mapping functions, unifying them under a general framework. This framework helped us to understand connections to existing formulations in the literature like softmax, spherical softmax and sparsemax. Our proposed probability mapping functions enabled us to provide explicit control over sparsity to achieve higher interpretability. These functions have closed-form solutions and sub-gradients can be computed easily. We have also proposed convex loss functions, which helped us to achieve better accuracies in the multilabel classification setting. Application of these formulations to compute sparse attention weights for NLP tasks also yielded improvements in addition to providing control to produce enhanced interpretability. As future work, we intend to apply these sparse attention formulations for efficient read and write operations of memory networks [19]. In addition, we would like to investigate application of these proposed sparse formulations in knowledge distillation and reinforcement learning settings.

**Acknowledgements**

We thank our colleagues in IBM, Abhijit Mishra, Disha Shrivastava, and Parag Jain for the numerous discussions and suggestions which helped in shaping this paper.

## Footnotes

[2]Micro-averaged $F_1$ score.

[3]Available at http://mulan.sourceforge.net/datasets-mlc.html

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
