[Supplementary Material]

# A  Supplementary Material

## A.1  Closed-Form Solution of Sparsegen

Formulation is given by:

$$\boldsymbol{p}^* = \text{sparsegen}(\boldsymbol{z}) = \underset{\boldsymbol{p} \in \Delta^{K-1}}{\text{argmin}} \{\|\boldsymbol{p} - g(\boldsymbol{z})\|_2^2 - \lambda\|\boldsymbol{p}\|_2^2\} \qquad (9)$$

The Lagrangian of the above formulation is:

$$\mathscr{L}(\boldsymbol{z}, \boldsymbol{\mu}, \tau) = \frac{1}{2}\|\boldsymbol{p} - g(\boldsymbol{z})\|_2^2 - \frac{\lambda}{2}\|\boldsymbol{p}\|_2^2 - \boldsymbol{\mu}^T\boldsymbol{p} + \tau(\mathbf{1}^T\boldsymbol{p} - 1) \qquad (10)$$

Defining the Karush-Kuhn-Tucker conditions with respect to optimal solutions $(\boldsymbol{p}^*, \boldsymbol{\mu}^*, \tau^*)$:

$$p_i^* - g_i(\boldsymbol{z}) - \lambda p_i^* - \mu_i^* + \tau^* = 0, \forall i \in [K], \qquad (11)$$

$$\mathbf{1}^T\boldsymbol{p}^* = 1, \boldsymbol{p}^* \geq \mathbf{0}, \boldsymbol{\mu}^* \geq \mathbf{0}, \qquad (12)$$

$$\mu_i^* p_i^* = 0, \forall i \in [K].(\text{complementary slackness}) \qquad (13)$$

From *Eq.*13, if we want $p_i^* > 0$ for certain $i \in [K]$, then we must have $\mu_i^* = 0$. This implies $p_i^* - g_i(\boldsymbol{z}) - \lambda p_i^* + \tau^* = 0$ (according to *Eq.* 11). This leads to $p_i^* = (g_i(\boldsymbol{z}) - \tau^*)/(1 - \lambda)$. For $i \notin S(\boldsymbol{z})$, where $p_i = 0$, we have $g_i(\boldsymbol{z}) - \tau^* \leq 0$ (as $\mu_i^* \geq 0$). From *Eq.* 12 we obtain $\sum_{j \in S(\boldsymbol{z})}(g_j(\boldsymbol{z}) - \tau^*) = 1 - \lambda$, which leads to $\tau^* = (\sum_{j \in S(\boldsymbol{z})} g_j(\boldsymbol{z}) - 1 + \lambda)/|S(\boldsymbol{z})|$.

**Proposition 0.1** *The closed-form solution of sparsegen is as follows ($\forall i \in [K]$):*

$$\rho_i(\boldsymbol{z}) = \text{sparsegen}_i(\boldsymbol{z}; g, \lambda) = \Big[\frac{g_i(\boldsymbol{z}) - \tau(\boldsymbol{z})}{1 - \lambda}\Big]_+, \qquad (14)$$

*where $\tau(\boldsymbol{z})$ is the threshold which makes $\sum_{j \in [K]} \rho_j(\boldsymbol{z}) = 1$. Let $g_{(1)}(\boldsymbol{z}) \geq g_{(2)}(\boldsymbol{z}) \cdots \geq g_{(K)}(\boldsymbol{z})$ be the sorted coordinates of $g(\boldsymbol{z})$. The cardinality of the support set $S(\boldsymbol{z})$ is given by $k(\boldsymbol{z}) := \max\{k \in [K] \mid 1 - \lambda + kg_{(k)}(\boldsymbol{z}) > \sum_{j \leq k} g_{(j)}(\boldsymbol{z})\}$. Then $\tau(\boldsymbol{z})$ can be obtained as,*

$$\tau(\boldsymbol{z}) = \frac{\sum_{j \leq k(\boldsymbol{z})} g_{(j)}(\boldsymbol{z}) - 1 + \lambda}{k(\boldsymbol{z})} = \frac{\sum_{j \in S(\boldsymbol{z})} g_j(\boldsymbol{z}) - 1 + \lambda}{|S(\boldsymbol{z})|},$$

## A.2  Special cases of Sparsegen

**Example 1**: $g(\boldsymbol{z}) = \boldsymbol{z}$ (**sparsegen-lin**):
When $g(\boldsymbol{z}) = \boldsymbol{z}$, sparsegen reduces to a regularized extension of sparsemax. This is translation invariant, has monotonicity and can work for full domain. It is also Lipschitz continuous with the Lipschitz constant being $1/(1 - \lambda)$. The visualization of this variant is shown in *Fig.*2.

**Example 2**: $g(\boldsymbol{z}) = \exp(\boldsymbol{z})$ (**sparsegen-exp**):
$\exp(\boldsymbol{z})$ here means element-wise exponentiation of $\boldsymbol{z}$, that is $g_i(\boldsymbol{z}) = \exp(z_i)$. For $\boldsymbol{z}$ where all $z_i > 0$, sparsegen-exp leads to sparser solutions than sparsemax.

Checking with properties from *Sec.*3.3, sparsegen-exp satisfies monotonicity as $\exp(z_i)$ is a monotonically increasing function. However, as $\exp(\boldsymbol{z})$ is not Lipschitz continuous, sparsegen-exp is not Lipschitz continuous.

It is interesting to note that sparsegen-exp reduces to softmax when $\lambda$ is dependent on $\boldsymbol{z}$ and equals $1 - \sum_{j \in [K]} e^{z_j}$, as it results in $\tau(\boldsymbol{z}) = 0$ and $S(\boldsymbol{z}) = [K]$ according to *Eq.*14.

**Example 3:** $g(\boldsymbol{z}) = \boldsymbol{z}^2$ (**sparsegen-sq**):

Unlike sparsegen-lin and sparsegen-exp, sparsegen-sq does not satisfy the monotonicity property as $g_i(\boldsymbol{z}) = z_i^2$ is not monotonic. Also sparsegen-sq has neither translation invariance nor scale invariance. Moreover, as $\boldsymbol{z}^2$ is not Lipschitz continuous, sparsegen-sq is not Lipschitz continuous. Additionally, when $\lambda$ depends on $\boldsymbol{z}$ and $\lambda = 1 - \sum_{j \in [K]} z_j^2$, sparsegen-sq reduces to spherical softmax.

**Example 4:** $g(\boldsymbol{z}) = log(\boldsymbol{z})$ (**sparsegen-log**):
$\log(\boldsymbol{z})$ here means element-wise natural logarithm of $\boldsymbol{z}$, that is $g_i(\boldsymbol{z}) = \log(z_i)$. This is scale invariant but not translation invariant. However, it is not defined for negative values or zero values of $z_i$.

### A.3 Derivation of Sparsecone

Let us project a point $\boldsymbol{z} = (z_1, \ldots, z_K) \in \mathbb{R}^K$ onto the simplex along the line connecting $\boldsymbol{z}$ with $\boldsymbol{q} = (-q, \ldots, -q) \in \mathbb{R}^K$. The equation of this line is $\alpha \boldsymbol{z} + (1 - \alpha)\boldsymbol{q}$. If the point of intersection of the line with the probability simplex (denoted by $\mathbf{1}^T \boldsymbol{z} = 1$) is represented as $\hat{\boldsymbol{z}} = \alpha^* \boldsymbol{z} + (1 - \alpha^*)\boldsymbol{q}$, then it should have the property: $\sum_i \hat{z}_i = \alpha^* \sum_i z_i - (1 - \alpha^*)Kq = 1$, which leads to $\alpha^* = (1 + Kq)/(\sum_i z_i + Kq)$. Also, $\hat{z}_i \geq 0$ condition must be satisfied, as it should lie on the probability simplex. Thus, the required point can be obtained by solving the following:

$$\boldsymbol{p}^* = \operatorname*{argmin}_{\boldsymbol{p} \in \Delta^{K-1}} \|\boldsymbol{p} - \hat{\boldsymbol{z}}\|_2^2 = \operatorname*{argmin}_{\boldsymbol{p} \in \Delta^{K-1}} \|\boldsymbol{p} - \alpha^* \boldsymbol{z} - (1 - \alpha^*)\boldsymbol{q}\|_2^2, \tag{15}$$

$$= \operatorname*{argmin}_{\boldsymbol{p} \in \Delta^{K-1}} \|\boldsymbol{p} - \alpha^* \boldsymbol{z}\|_2^2 = \text{sparsemax}(\alpha^* \boldsymbol{z}), \tag{16}$$

$$\alpha^* = \frac{1 + Kq}{\sum_{i \in [K]} z_i + Kq} \tag{17}$$

As $\alpha^*$ is a function of $\boldsymbol{z}$, it is denoted by $\alpha(\boldsymbol{z})$. The formulation in *Eq.*15 is equivalent to *Eq.*16, utilizing the translation invariance property of sparsemax as $(1 - \alpha^*)\boldsymbol{q}$ is a constant term for a particular $\boldsymbol{z}$. Thus, we can say sparsecone can be obtained by sparsemax$(\alpha(\boldsymbol{z})\boldsymbol{z})$, where $\alpha(\boldsymbol{z})$ is given by *Eq.*17.

### A.4 Derivation of Sparsehourglass

The above formulation fails for $\boldsymbol{z}$ such that $\sum_i z_i < -Kq$. For example, it fails for $\boldsymbol{z} = (-2, -1)$, where $\boldsymbol{q} = (-1, -1)$ (the solution will give greater probability mass to -2 compared to -1), that is $q = 1$. To make it work for such a $\boldsymbol{z}$, we can find a *mirror point* $\tilde{\boldsymbol{z}}$ satisfying the following properties : (1) $\sum_i \tilde{z}_i = -\sum_i z_i$, and (2) $\tilde{z}_i - \tilde{z}_j = z_i - z_j \forall (i, j)$, and (3) sparsehourglass$(\boldsymbol{z}) = $ sparsecone$(\tilde{\boldsymbol{z}})$. Let sparsecone$(\tilde{\boldsymbol{z}})$ be given by sparsemax$(\alpha(\tilde{\boldsymbol{z}})\tilde{\boldsymbol{z}})$ (as seen earlier) and sparsehourglass$(\boldsymbol{z})$ be defined by sparsemax$(\alpha^* \boldsymbol{z})$. Can we find $\alpha^*$ which satisfies the mirror point properties above? Property (3) is true iff $\alpha(\tilde{\boldsymbol{z}})(\tilde{z}_i - \tilde{z}_j) = \alpha^*(z_i - z_j)\forall (i, j)$. Using property (2) we get, $\alpha(\tilde{\boldsymbol{z}}) = \alpha^*$. From definition of sparsecone, $\alpha(\tilde{\boldsymbol{z}}) = (1 + Kq)/(\sum_i \tilde{z}_i + Kq)$. Using property (1), we get $\alpha^* = \alpha(\tilde{\boldsymbol{z}}) = (1 + Kq)/(\sum_i \tilde{z}_i + Kq) = (1 + Kq)/(-\sum_i z_i + Kq)$. As $\alpha^*$ is a function of $\boldsymbol{z}$, let it be represented as $\hat{\alpha}(\boldsymbol{z})$. Thus, **sparsehourglass** is defined as follows:

$$\boldsymbol{p}^* = \text{sparsehourglass}(\boldsymbol{z}) = \operatorname*{argmin}_{\boldsymbol{p} \in \Delta^{K-1}} \|\boldsymbol{p} - \hat{\alpha}(\boldsymbol{z})\boldsymbol{z}\|_2^2 \tag{18}$$

$$= \text{sparsemax}(\hat{\alpha}(\boldsymbol{z})\boldsymbol{z}) \tag{19}$$

$$\hat{\alpha}(\boldsymbol{z}) = \frac{1 + Kq}{|\sum_{i \in [K]} z_i| + Kq} \tag{20}$$

### A.5 Proof for Lipschitz constant of Sparsehourglass

Lipschitz constant of composition of functions is bounded above by the product of Lipschitz constants of functions in the composition. Applying this principle on *Eq.*19 we find an upper bound for the Lipschitz constant of sparsehg (denoted as $L_{\text{sparsehg}}$) as the product of Lipschitz constant of sparsemax (denoted as $L_{\text{sparsemax}}$) and Lipschitz constant of $g(\boldsymbol{z}) = \hat{\alpha}(\boldsymbol{z})\boldsymbol{z}$ (which we denote as $L_g$).

$$L_{\text{sparsehg}} \leq L_{\text{sparsemax}} \times L_g \tag{21}$$

We use the property that Lipschitz constant of a function is the largest matrix norm of Jacobian of that function. For sparsemax that would be,

$$L_{\text{sparsemax}} = \operatorname*{argmax}_{\boldsymbol{x}, \boldsymbol{z} \in \mathbb{R}^K} \frac{\|J_{\text{sparsemax}}(\boldsymbol{z})\boldsymbol{x}\|_2}{\|\boldsymbol{x}\|_2}$$

where, $J_{\text{sparsemax}}(\boldsymbol{z})$ is the Jacobian of sparsemax at given value $\boldsymbol{z}$. From the term given in *Sec.*3, $J_{\text{sparsemax}}(\boldsymbol{z}) = \left[\text{Diag}(\boldsymbol{s}) - \frac{\boldsymbol{s}\boldsymbol{s}^T}{|S(\boldsymbol{z})|}\right]$ whose matrix norm is 1. We use the property that, for any symmetric matrix, matrix norm is the largest absolute eigenvalue, and the eigenvalues of $J_{\text{sparsemax}}(\boldsymbol{z})$ are 0 and 1. Hence, $L_{\text{sparsemax}} = 1$.

For $g(\boldsymbol{z}) = \hat{\alpha}(\boldsymbol{z})\boldsymbol{z}$, let us look at the Jacobian of $g(\boldsymbol{z})$, which is given as

$$J_g(\boldsymbol{z}) = \hat{\alpha}(\boldsymbol{z})\mathbf{I} - \frac{\hat{\alpha}(\boldsymbol{z})\operatorname{sgn}(\sum_i z_i)}{|\sum_i z_i| + Kq} \begin{bmatrix} z_1 & z_1 & \dots & z_1 \\ z_2 & z_2 & \dots & z_2 \\ \vdots & \ddots & & \\ z_K & & & z_K \end{bmatrix}, \tag{22}$$

where $\mathbf{I}$ denotes identity matrix. Eigen values of $J_g(\boldsymbol{z})$ are $\hat{\alpha}(\boldsymbol{z})$ and $\hat{\alpha}(\boldsymbol{z})(1 - \frac{|\sum z_i|}{|\sum z_i| + Kq})$, and the largest among them is clearly $\hat{\alpha}(\boldsymbol{z})$. As $\hat{\alpha}(\boldsymbol{z}) > 0$, all the eigen values of $J_g(\boldsymbol{z})$ are greater than 0, making $J_g(\boldsymbol{z})$ a positive definite matrix. We now use the property that for any positive definite matrix, matrix norm is the largest eigenvalue. From *Eq.*20, we see that the largest eigen value of $J_g(\boldsymbol{z})$ is $\hat{\alpha}(\boldsymbol{z})$ which assumes the largest value when $\sum_i z_i = 0$. The highest value of $\hat{\alpha}(\boldsymbol{z})$ is $1 + \frac{1}{Kq}$; therefore, $L_g = 1 + \frac{1}{Kq}$. Thus,

$$L_{\text{sparsehg}} \leq L_{\text{sparsemax}} \times L_g = 1 \times (1 + \frac{1}{Kq}) = 1 + \frac{1}{Kq}$$

It turns out this is also a tight bound; hence, $L_{\text{sparsehg}} = 1 + \frac{1}{Kq}$.

## A.6 Synthetic dataset creation

We use scikit-learn for generating synthetic datasets with 5000 instances split across train (0.5), val (0.2) and test (0.3). Each instance is a multilabel document with a sequence of words. Vocabulary size and number of labels $K$ are fixed to 10. Each instance is generated as follows: We draw number of true labels $N \in \{1, \dots, K\}$ from either a discrete uniform distribution. Then we sample $N$ labels from $\{1, \dots, K\}$ without replacement and sample $L$ number of words (document length) from the mixture of sampled label-specific distribution over words.

## A.7 More results on multilabel classification experiments

### A.7.1 Synthetic Dataset

*Fig.*6 shows JSD between normalized true label and predicted distribution. Though sparsemax+huber is performing overall the best, in the case when number of labels is small we could see sparsehg+hinge and sparsemax+hinge performing as good as sparsemax+huber. In general we could see that sparsehg+hinge is performing better than sparsemax+hinge, which in turn is better than softmax+log.

(a) Varying mean #labels      (b) Varying range #labels      (c) Varying document length

Figure 6: JSD on multilabel classification synthetic dataset

### A.7.2 Real Datasets

Table 3: F-score on three benchmark multilabel datasets

| Activation + Loss | SCENE | EMOTIONS | BIRDS |
|---|---|---|---|
| softmax+log | **0.71** | **0.65** | **0.43** |
| sparsemax+huber | 0.70 | 0.63 | 0.42 |
| sparsemax+hinge | 0.68 | **0.65** | 0.41 |
| sparsehg+hinge | 0.69 | **0.65** | 0.41 |

## A.8 Controlled Sparse Attention

Here we discuss attention mechanism with respect to encoder-decoder architecture as illustrated in *Fig.* 7. Lets say we have a sequence of inputs (possibly sequence of words) and their corresponding word vectors sequence be $\boldsymbol{X} = (\boldsymbol{x}_1, \ldots, \boldsymbol{x}_K)$. The task will be to produce a sequence of words $\boldsymbol{Y} = (y_1, \ldots, y_L)$ as output. Here an encoder RNN encodes the input sequence into a sequence of hidden state vectors $\boldsymbol{H} = (\boldsymbol{h}_1, \ldots, \boldsymbol{h}_K)$. A decoder RNN generates its sequence of hidden states $\boldsymbol{S} = (\boldsymbol{s}_1, \ldots, \boldsymbol{s}_L)$ by considering different $\boldsymbol{h}_i$ while generating different $\boldsymbol{s}_j$. This is known as *hard attention*. This approach is known to be non-differentiable as it is based on a sampling based approach [6]. As an alternative solution, a softer version (called *soft attention*) is more popularly used. It involves generating a score vector $\boldsymbol{z}_t = (z_{t1}, \ldots, z_{tK})$ based on relevance of encoder hidden states $\boldsymbol{H}$ with respect to current decoder state vector $\boldsymbol{s}_t$ using *Eq.*23. The score vector is then transformed to a probability distribution using the *Eq.*24. Considering the attention model parameters $[\boldsymbol{W}_s, \boldsymbol{W}_h, \boldsymbol{v}]$, we define the following:

$$z_{ti} = \boldsymbol{v}^T \tanh(\boldsymbol{W}_s \boldsymbol{s}_t + \boldsymbol{W}_h \boldsymbol{h}_i). \tag{23}$$

$$\boldsymbol{p}_t = softmax(\boldsymbol{z}_t) \tag{24}$$

Following the approach by [14], we replace *Eq.*24 with our formulations, namely, sparsegen-lin and sparsehourglass. This enables us to apply explicit control over the attention weights (with respect to sparsity and tradeoff between scale and translation invariance) produced for the sequence to sequence architecture. This falls in the *sparse attention* paradigm as proposed in the earlier work. Unlike hard attention, these formulations lead to easy computation of sub-gradients, thus enabling backpropagation in this encoder-decoder architecture composed RNNs. This approach can also be extended to other forms of non-sequential input (like images) as long as we can compute the score vector $\boldsymbol{z}_t$ from them, possibly using a different encoder like CNN instead of RNN.

Figure 7: Encoder Decoder with Sparse Attention

### A.8.1 Neural Machine Translation

The first task we consider for our experiments is neural machine translation. Our aim is to see how our techniques compare with softmax and sparsemax with respect BLEU metric and interpretability. We consider the french-english language pair from the NMT-Benchmark project[4] and perform experiments both ways using controlled sparse attention. The dataset has around 1M parallel training instances along with equal validation and test sets of 1K parallel instances.

We find from our experiments (refer *Table.*2) that sparsegen-lin surpasses the BLEU scores of softmax and sparsemax for FR-EN translation, whereas the other formulations yield comparable performance. On the other hand, for EN-FR translation, softmax is still better than others. Quantitatively, these metrics show that adding explicit controls do not come at the cost of accuracy. In addition, it is encouraging to see (refer *Fig.*8) that increasing $\lambda$ for sparsegen-lin leads to crisper and hence more interpretable attention heatmaps (the lesser number of activated columns per row the better it is).

Figure 8: The attention heatmaps for softmax, sparsemax and sparsegen-lin ($\lambda = 0.75$) in translation (FR-EN) task.

Figure 9: The attention heatmaps for softmax, sparsemax and sparsegen-lin ($\lambda = 0.75$) in summarization (Gigaword) task.

### A.8.2 Abstractive Summarization

We next perform our experiments on abstractive summarization datasets like Gigaword, DUC2003 & DUC2004[5]. This dataset consists of pairs of sentences where the task is to generate the second sentence (news headline) as a summary of the larger first sentence. We trained our models on nearly 4M training pairs of Gigaword and validated on 190K pairs. Then we reported the test scores according to ROUGE metrics on 2K test instances of Gigaword, 624 instances of DUC2003 and 500 instances of DUC2004.

The results in *Table.*2 show that sparsegen-lin stands out in performance amongst other formulations with the other formulations closely following and comparable to softmax and sparsemax. It is also encouraging to see that all the models trained on Gigaword generalizes well on other datasets DUC2003 and DUC2004. Here again the $\lambda$ control leads to more interpretable attention heatmaps as shown in *Fig.*9.

## Footnotes

[4]http://scorer.nmt-benchmark.net/

[5]https://github.com/harvardnlp/sent-summary