[Reviews · NeurIPS 2018]

Reviewer 1



Note: in view of other related papers pointed out during the discussion process, I have adjusted the rating to reflect concerns over the contribution of this work. This submission presents two new methods, namely sparseflex and sparsehourglass, for mapping input vectors to the probability simplex set (unit sum vectors in the positive orthant). The main motivation is to improve the popular softmax function to induce sparse output vectors, as advocated by the sparsemax function. To this end, a general optimization framework (sparsegen) to the design of such probably mapping functions is proposed, by minimizing the mismatch to a transformation of the input penalized by negative Euclidean norm. Interestingly, it turns out that the sparsegen is equivalent to sparsemax, and it is possible to recover various existing mapping functions by choosing different transformation g(.) and penalization coefficient lambda. Two special realizations of sparsegen - sparseflex and sparsehourglass - have been introduced, along with the discussions on desirable properties for probably mapping functions. Last, the present submission also discusses the choice of loss function in multilabel classification for using the sparsegen framework. The presented work is analytically interesting and the proposed mapping functions have demonstrated practical values in experimental results. Some comments are listed here: - The introduction of sparseflex is missing in the main submission. As one of the two proposed mapping functions, the analytical form of sparseflex should be included in the submission. - Some claims are not formally corroborated. For example, in the introduction of multilabel classification (Sec 4), the issue of KL divergence needs reference papers to justify. - As both sparseflex and sparsehourglass stem from special settings of sparsegen, their parameter selections can be more explained especially for real applications such as speech attention. It is expected that their performance depends on the sparse regions as shown in Figs 2 and 3. More discussions on the rationals behind their settings would be nice.

Reviewer 2



The paper proposes a unified framework of probability mapping functions that induces softmax, sum-normalization, spherical softmax, and sparsemax as special cases. Two novel sparse formulations are proposed under the framework that allows for a control of desired sparsity level. Experiments on synthetic and some standard NLP tasks demonstrate the effectiveness of the new mapping functions. The paper is clear and well written. The proposed Sparsegen framework is novel that not only houses a number of existing probability mapping functions but also generates two new ones showing positive results on standard NLP tasks. It would be interesting to see whether the new functions are able to help improve/get close faster to the state-of-the-art performance on these tasks, though it may be hard to sort out the contribution as attention is just one component of the network.

Reviewer 3



Summary ------- This paper introduces several variants of the sparsemax mapping. Experiments are conducted on multi-label classification and attention mechanisms for neural machine translation / summarization. With some exceptions (see below), the paper is overall clearly written. The authors put great effort in explaining their ideas with figures. Experiments on two different tasks (multi-label classification and NMT) is also very positive. That said, the technical contribution seems very limited. The proposed "unifying" framework is just sparsemax with different inputs. As a result, many of the proofs in the Appendix are just existing proofs with different inputs. Sparseflex is just sparsemax with a regularization parameter, which was already proposed in [16]. Sparsecone and sparsehourglass feel quite adhoc and it is not clear why they should be any better than sparsemax. In the experiments, I am surprised sparsemax performed so much worse than sparseflex (sometimes up to 10 point difference!). As I explain below, sparsemax and sparseflex should perform exactly the same in theory. Therefore, I suspect there are issues in the empirical validation. Overall, I feel the paper is quite below the acceptance bar for NIPS. Major comments -------------- * I am sorry to be a bit blunt but the current title is really bad. I would really avoid bad analogies or bad puns in a title. My recommendation is to keep titles formal and informative. How about just "On Sparse Alternatives to Softmax"? * [16] is a very relevant work and I was surprised not to see it mentioned until the experiment section. I think it deserves a mention in the introduction, since it also provides a unifying framework. * Theorem 1 is a trivial result. I wouldn't call it a theorem. * Section 3.2: So is sparseflex just sparsemax with a regularization parameter? The fact that tunning lambda allows to control sparsity was already mentioned in [16]. * Introducing too many new names (sparseflex, sparsecone, etc) is not a good idea, as it confuses the reader. My recommendation is to pick only one or two proposed methods. You don't need to introduce new names for every special case. * I am surprised that sparsemax performs so much worse than sparseflex in Figure 5 and 6. Indeed, if the vectors of scores z is produced by a linear model, i.e., z = W x, the regularization parameter can be absorbed into W as is obvious from Eq. (2). Given this fact, how do you explain why sparseflex performs so much better? * Appendices A.1 to A.4 are essentially the same proof again and again with different inputs. Since sparsegen reduces to the projection on the simplex, there is no need for all this... * For multi-label classification, experiments on real data would be welcome. Also it'd be great to report the average sparsity per sample on the test set. Detailed comments ----------------- * Line 62: Z_+ is commonly denoted N. * Line 63: {a, b, c} usually defines the elements of a set, not a vector * There are many works tackling the projection onto the simplex before [15]. It would great to give them proper credit. Some examples: P.Brucker. An O(n) algorithm for quadratic knapsack problems. Operations Research Letters,3(3):163–166, 1984. C. Michelot. A finite algorithm for finding the projection of a point onto the canonical simplex of Rn. Journal of Optimization Theory and Applications, 50(1):195–200, 1986. * Line 209: multiple labels can be true => multiple labels are allowed ---------- > However, A.3 follows directly from A.1 and can be removed. I still think that A.1 is basically a known result since Eq. (6) (supp) can be reduced to the projection on the simplex. > [On empirical evaluation] Thank you for the clarification between L_sparsemax and L_sparseflex. This addressed my concern about Fig. 5.

Reviewer 4



This paper introduces new sparsity inducing design paradigm for probability distributions that generalizes existing transformations and have desirable properties such as (approx) scale/translate invariance, lipschitz continuity, monotonicity, full domain etc. The paper is well-motivated and written, with ample examples explaining the behavior of the loss functions and experiments. The paper also describes an application to multilabel classification, and provides closed form functional updates to train based on the proposed framework. The resulting loss function is also convex; the authors also draw comparison to hinge loss. I think this paper should draw many discussions at nips. There is no theoretical consistency studies, and while the empirical evaluations could have been expanded to more applications, nevertheless this is a strong paper imho. For completeness, can the authors also include the derivation of the Lipschitz continuity of 1+1/Kq. I would suggest reporting of time taken to train, since constraint violation type losses are slower to train on. Also, it should be easy to fashion a smoothened version that can be formulated that is not as "strict" but at the same time does not require to worry about subgradients. Is there an intuitive explanation of why sparsehg does not seem to perform as well ? It trades off for more area of the domain to have a sparse representation, is it over-regularizing ?